# The Impact of Infectious Diseases on Clinical Characteristics and Immunological Correlations in Pediatric Henoch–Schönlein Purpura: A Five-Year Retrospective Study

**DOI:** 10.3390/biomedicines13010113

**Published:** 2025-01-06

**Authors:** Sînziana Oprițescu, Gabriela Viorela Nițescu, Mihaela Golumbeanu, Dora Boghițoiu, Elena Iuliana Ioniță, Diana-Andreea Ușurelu, Cristian Lucaci, Adriana Negoiță, Elena Moroșan

**Affiliations:** 1Discipline of Clinical Laboratory and Food Safety, Faculty of Pharmacy, “Carol Davila” University of Medicine and Pharmacy, 6 Traian Vuia Street, 020945 Bucharest, Romania; sinziana.opritescu@drd.umfcd.ro (S.O.); elena.morosan@umfcd.ro (E.M.); 2Discipline of Pediatrics, Faculty of Dentistry, “Carol Davila” University of Medicine and Pharmacy, 020021 Bucharest, Romania; 3“Grigore Alexandrescu” Clinical Emergency Hospital for Children, 017443 Bucharest, Romania; 4Discipline of Pharmacognosy, Phytochemistry and Phytotherapy, Faculty of Pharmacy, “Carol Davila”, University of Medicine and Pharmacy, 6 Traian Vuia Street, 020945 Bucharest, Romania

**Keywords:** Henoch–Schönlein purpura, IgA vasculitis, allergic vasculitis, infectious diseases

## Abstract

**Background/Objectives:** Immunoglobulin A (IgA) vasculitis (IgAV), classically known as Henoch–Schönlein purpura (HSP), is a type of nonthrombocytopenic small-vessel vasculitis. HSP is the most frequent kind of systemic vasculitis in children, characterized by purpura, arthritis or arthralgia, gastrointestinal pain, and kidney dysfunction. The aim of our research was to investigate and observe the clinical characteristics of children diagnosed with HSP and to explore the correlation between infectious diseases and HSP. Furthermore, this retrospective study considered other factors, such as demographic characteristics (sex, area/environment, and age), and their effect on the pediatric population diagnosed with HSP. **Methods:** To answer this question, we conducted a five-year hospital-based retrospective study that included 144 hospitalized children who were diagnosed with HSP during hospitalization. Measurements of immunological panels (IgA, IgM, IgG, and IgE), C3, C4, C-reactive protein, fibrinogen, and hematite sedimentation rate (VSH) determined using blood samples revealed that there is a strong correlation between the elements of the immunological panel and the HSP manifestations. **Results:** Additionally, elevated IgG and normal IgA serum levels in pediatric HSP patients are strongly associated with infectious diseases. **Conclusions**: Notably, patients with infectious diseases exhibited high IgG and normal IgA serum levels post-treatment and a higher risk of relapses.

## 1. Introduction

### 1.1. Henoch–Schönlein Purpura Incidence and Clinical Manifestations

Immunoglobulin A (IgA) vasculitis (IgAV), classically known as Henoch–Schönlein purpura (HSP), is a type of nonthrombocytopenic small-vessel vasculitis [1]. HSP is the most frequent kind of systemic vasculitis in children, characterized by purpuric lesions (on the face, ears, extremities, and scrotum), arthritis or arthralgia, gastrointestinal pain, and kidney dysfunction [2,3]. Systemic IgAV can occur at any age; however, it most commonly affects children aged 3 to 15 years. In total, 90% of childhood-onset diseases develop before the age of ten years [1,4,5]. The incidence of the pathology ranges from 3 to 27 cases per 100,000 children [6,7]; thus, it is relatively uncommon among infants [6].

Although the precise pathogenesis of HSP is undetermined, pathological and laboratory findings, such as a vascular deposition of the IgA-dominant immune complex, the infiltration of small blood vessels with polymorphonuclear leukocytes, the presence of leukocyteoclasia, increased serum IgA, and proinflammatory cytokines, indicate that HSP is an immune-mediated disease [8,9].

### 1.2. Immunoglobulin A’s Role and Characterization in Pathogenesis

IgA, a member of the human immunoglobulin family, serves as an essential component of mucosal defense, primarily by neutralizing bacterial products, agglutinating microorganisms, and interfering with bacterial movement [10,11,12]. IgA is produced in greater quantities than all other immunoglobulin classes combined due to its high mucosal production and short half-life of 5–6 days [13,14]. Increased IgA synthesis could be linked to the antigen exposure processed by the mucosal immune system. Bacteria, viruses, or parasitic agents were hypothesized to initiate the HSP disease in genetically predisposed individuals, but causal agents and factors have yet to be found [15,16]. During the acute stage of HSP, a rise in IgA may indicate aberrant immunological responses; IgA1 accounts for over 90% of IgA in the blood, while IgA2 accounts for less than 10% [10,17,18].

### 1.3. Potential Etiological Factors for Henoch–Schönlein Purpura

As mentioned before, the pathophysiology of HSP is poorly known; however, numerous medical professionals are aware that it is usually associated with a variety of viral disorders, as well as exposure to insect bites, immunizations, medicines, or food allergies [16,19,20,21]. Although the etiology of the disease remains unknown, it is obvious that the IgA system plays an important part in the pathogenesis [22]. A variety of triggers have been proposed, including infections and medicines [3,23,24]. Moreover, the hallmark pathogenic feature (the deposition of IgA-containing immune complexes within tiny artery walls and renal mesangium) suggests the involvement of mucosal infections [25,26,27]. IgAV has a seasonal tendency, with fewer instances occurring during the summer months, confirming the hypothesis that viral precipitants cause the start of this disease [28,29].

Our research aimed to evaluate clinical data from pediatric patients diagnosed with HSP and investigate the association between infectious diseases and the described pathology. Additionally, our retrospective study examined demographic factors (including sex, area/environment, and age) and their impact on the pediatric HSP population. This study will emphasize the key clinical aspects of HSP in the studied population.

## 2. Materials and Methods

### 2.1. Study Design

The clinical research was conducted as a hospital-based retrospective study, including 144 children (1 to 18 years of age), all of whom were patients admitted to the “Grigore Alexandrescu” Emergency Clinical Hospital for Children in Bucharest. Given its status as an emergency and university hospital, patients were referred from across the country. We searched the electronic medical database of the hospital for relevant records from 1 January 2017 to 31 December 2023. All children were previously referred for evaluation due to HSP. The children diagnosed with HSP were classified according to disease severity using the European League Against Rheumatism (EULAR), the Pediatric Rheumatology International Trials Organization (PRINTO), and the Pediatric Rheumatology European Society (PRES) classification criteria for HSP [6].

The samples were obtained from venous blood for the quantitative measurement of the immunological panel (IgA, IgM, IgG, and IgE), C3, C4, C-reactive protein, fibrinogen, and VSH. An anticoagulant-free vacutainer with or without separating gel served as the collecting container. Levels of immunological panel elements were measured with an enzyme-linked immunosorbent assay (ELISA) [30,31,32,33,34,35,36].

The collected patient files were entered into a database. The hospital database was used to collect information such as age, gender, environment, immunological panel (IgA, IgM, IgG, and IgE), C3, C4, C-reactive protein, fibrinogen, and VSH clinical diagnosis and other associated diagnosis. The data were statistically evaluated, and the statistical tests applied are detailed in the ‘Descriptive Analysis of the Patients’ Series’ section (Section 2.2).

After the diagnosis of HSP, the assessment of disease severity (mild, moderate, or severe) was determined in accordance with the guidelines set forth by the EULARPRINTO/PRES. These organizations have delineated the current diagnostic criteria for systemic IgA vasculitis (IgAV) [6,24,37]. In a pediatric patient presenting with purpura, characterized as round or oval and retiform, predominantly on the lower limbs, the diagnosis is confirmed if at least one of the following four criteria is met: (1) abdominal pain, (2) histologically confirmed IgA deposits, (3) arthritis or arthralgia, or (4) renal impairment [6,24,37].

### 2.2. Descriptive Analysis of the Patients’ Series

#### 2.2.1. Inclusion Criteria

The patients included in the study were of both sexes, with an age range between 1 and 18 years, admitted to the Pediatric Clinic of “Grigore Alexandrescu” Emergency Clinical Hospital for Children in Bucharest. The children were previously diagnosed with HSP and had their immunological panel (IgA, IgM, IgG, and IgE), C3, C4, C-reactive protein, fibrinogen, and VSH tested. All the patients included in the study presented an infectious disease during the last 6 months before hospitalization. To ensure the integrity of the study, only patients without a history of COVID-19 infection were included, due to the complex treatment protocols associated with the disease.

We defined “mild” symptoms in children with HSP as rash/purpura (small, reddish-purple spots that typically appear on the lower legs and gluteal region), joint pain (mild pain and swelling, usually in the knees and ankles), and digestive symptoms (occasional mild abdominal pain that has no significant effect on appetite or daily routines).

We defined “moderate” symptoms in children with HSP as those exhibiting rash/purpura (more extensive purpura that may spread to the arms, face, and trunk, causing moderate discomfort), joint pain (more pronounced pain and swelling in multiple joints, potentially limiting mobility and daily activities), digestive symptoms (frequent abdominal pain, nausea, and occasional vomiting, which can impact appetite and daily routines), and kidney involvement (presence of blood or protein in the urine, detectable through urine tests, indicating moderate kidney involvement).

We identified “severe” symptoms in children with HSP as those presenting with rash/purpura (extensive and severe purpura that can be painful and cover large areas of the body), joint pain (severe pain and swelling in multiple joints, significantly impairing mobility and daily activities), digestive symptoms (severe abdominal pain, persistent vomiting, and bloody stools, which can lead to dehydration and require medical intervention), and kidney involvement (significant kidney damage, leading to persistent blood or protein in the urine, and potentially progressing to chronic kidney disease).

#### 2.2.2. Exclusion Criteria

The following patients were excluded from the study: patients below 1 year of age, patients not previously diagnosed with HSP, and patients who had no immunological panel test performed.

#### 2.2.3. Statistical Analysis of Data

The data were statistically analyzed using IBM’s Statistical Analysis Software Package (SPSS) version 29 (2022) and Microsoft Excel 2016 (Redmon, WA, USA). The investigation included descriptive statistics, tests to assess normal distribution (Kolmogorov–Smirnov and Shapiro–Wilks), tests to compare quantitative indicators in different groups (comparison of means), correlation analyses (Pearson correlation coefficient, PCC), ROC curves, positive predictive value (PPV), negative predictive value (NPV), and sensitivity and specificity. The chosen significance level was α = 0.05 and 0.01 for PCC. Thus, if the significance level is not reached for values of *p* < α, the null hypothesis is rejected.

## 3. Results

### 3.1. Descriptive Statistics

A retrospective hospital-based study was conducted over a five-year period, during which 144 patients were diagnosed with HSP. Of the 144 total children, 87 (60.4%) were males and 57 (39.6%) were females. The boys were slightly younger than the girls, with a mean age of 7.18 ± 3.907 years vs. 8.00 ± 4.044 years, respectively. Overall, the mean age of the group was 7.51 ± 3.96 years. According to the Kolmogorov–Smirnov and Shapiro–Wilk statistical tests, neither gender had a normal age distribution (*p* < 0.01 for both sexes). The median age of the total cohort was 7 years, with females at 7 years and males at 6.78 years. The interquartile range (IQR) for the overall group was 6 years, with the females’ IQR at 6 and the males’ IQR at 5. There was a 1.57:1 ratio between the urban and rural patients (urban/rural, 88:56).

Among the 144 patients included in the study, a significant proportion (65.27%) exhibited a moderate disease status, particularly within the urban patient group, while the smallest group of patients (9.02%) exhibited the mildest symptoms. Notably, severe manifestations were observed in 25.69% of patients, with a nearly equal distribution between urban and rural areas. Among patients diagnosed with severe HSP, 63.8% were male. Figure 1 illustrates the distribution of patients with varying severity levels of HSP alongside the environmental distribution.

The clinical study documented the date of hospitalization, revealing that during autumn, the highest patient count across all seasons was recorded (41 patients, 28.47%). In contrast, the number of hospitalized patients during the winter and spring was nearly equal (38 patients during the winter and 39 patients during the spring). Interestingly, summer had the lowest hospitalization rate (26 patients, 18.05%), with no patients exhibiting mild-severity symptoms during that season. Figure 2 illustrates the distribution of patients according to season and the severity of HSP.

### 3.2. Immunologic Panel Results

We investigated potential correlations between IgA serum levels and immunological panel elements within a patient cohort. The immunological panel analysis included, besides IgA, immunoglobulin G (IgG), immunoglobulin M (IgM), and immunoglobulin E (IgE). Moreover, we included in the Principal Component Analysis (PCA) the inflammation markers such as complement 3 (C3), complement 4 (C4), fibrinogen, sedimentation velocity of hematite (VSH), and C-reactive protein (PCR). The test results indicated underlying correlations that exist in the set of variables; the Kaiser–Meyer–Olkin (KMO) measure of sampling adequacy was 0.636, and Bartlett’s test of sphericity yielded a chi-square value of 188.711 with 36 degrees of freedom (*p* < 0.001). According to the test results, there are four extracted components that satisfy the condition of a correlation coefficient greater than or equal to 1. The four components extracted are all the immunological panel elements. These results are presented in Figure 3 and Table A1.

In the study, the Pearson correlation coefficient (PCC) was applied to explore the potential correlations among elements in the immunological panel. Our findings indicate moderate positive correlations between IgA and IgG (PCC = 0.238, *p*-value = 0.01), as well as between IgA and IgM (PCC = 0.443, *p*-value = 0.01). Additionally, a weaker positive correlation exists between IgG and IgM (PCC = 0.222, *p*-value = 0.01).

To investigate potential immunological correlations, two blood samples were collected from each patient enrolled in the study. The initial sample was taken upon hospital admission, while the second sample was collected 30 days post-discharge. Furthermore, the patients were divided into two groups: the first group comprised individuals with an infectious disease as a secondary diagnosis, while the second group consisted of those without any infectious diagnoses upon admission. Analysis of the immunological panel revealed that 33.33% of patients exhibited elevated IgA levels, 34.02% exhibited elevated IgM levels, and only 14.58% exhibited elevated IgG levels during the initial phase of the study.

A paired sample *t*-test was conducted to determine whether there was a statistically significant difference between the means of the immunoglobulin serum levels measured during the first and second phases of sample collection. The results indicated a significant difference between the means of each group. The findings of the paired samples *t*-test reveal significant variations in immunoglobulin serum levels between the first and second stages of sample collection. For each pair of immunological parameters (IgA, IgM, and IgG), the mean differences were substantial, with *p*-values less than 0.05, suggesting that the changes in immunoglobulin levels over time are statistically significant. The confidence intervals for these differences do not include zero, further confirming the significance of these results. While all pairs attained statistical significance, a considerable disparity existed between the difference of means of pairs 1 and 2. Pair 1 comprised the IgA serum levels in the cohort with an infectious disease as a secondary diagnosis, while Pair 2 included the same parameter for patients without an infectious disease as a secondary diagnosis. The difference of means result for Pair 1 was notably low (0.04517), in contrast to Pair 2, which exhibited a significantly higher difference of means (0.23812). For Pair 3 and Pair 4, which measure the IgM serum levels in patients with and without a subsequent infectious disease diagnosis, the difference of means did not demonstrate significant differences. The difference of means for Pair 3 was 0.19017, while the difference of means for Pair 4 was 0.10587. The concluding pairs, Pair 5 and Pair 6, evaluate the IgG serum concentrations in patients with and without a future diagnosis of infectious disease. Similar to the initial pairs, there was a significant difference between the difference of means. The difference of means for Pair 6 were notably low (0.15312), in contrast to Pair 5, which exhibited a significantly higher mean (−2.54373). Additionally, for Pair 5, the mean during the first phase of sample collection was significantly lower compared to the mean during the second phase of sample collection. These results are detailed in Figure 4 and Table A2.

The Independent Samples Test results reveal substantial disparities between the means of the compared groups. Levene’s Test for Equality of Variances shows that the variances are equal, as the significance values are greater than 0.05. The *t*-test for equality of means reveals that the differences in means are statistically significant, with *p*-values less than 0.05. The mean differences and their corresponding 95% confidence intervals indicate that the true mean differences are unlikely to be zero, confirming the significance of the results. This suggests that the groups have distinct immunoglobulin serum levels, highlighting the impact of the presence or absence of an infectious disease on these levels. These results are detailed in Figure 4 and Table A3.

### 3.3. Patients with Infectious Diseases as Secondary Diagnoses

In this study, we used the Receiver Operating Characteristic (ROC) curve to assess the diagnostic performance of serum IgA levels in distinguishing between patients who had experienced infectious diseases as a secondary diagnosis during admission and those who had not. The ROC curve results indicated that IgA levels achieved statistical significance, with an AUC of 0.855 and a standard error of 0.031, demonstrating a strong discriminative capacity for diagnosing normal IgA levels in patients with infections, as detailed in Table 1 and in the graphic representation presented in Figure 5. Moreover, the Youden’s index equals 0.567, with a cutoff value for the ROC curve of 2.000 g/L, suggesting that the test has a reasonably good ability to distinguish between positive and negative cases. The ROC curve analysis was conducted on a cohort of 141 patients included in the study; the remaining patients did not undergo testing for their total IgA serum levels. The patients who did not undergo testing for total IgA serum levels presented very mild symptoms; thus, the hospital protocol did not include the IgA serum levels. Patients with normal IgA levels and infection as secondary clinical diagnoses were classified as true positives (43 patients), whereas those with high IgA levels but no clinical infectious disease diagnoses were classified as false negatives (31 patients). True negatives included patients with normal IgA levels and no infection as secondary clinical diagnoses (50 patients), and false positives comprised patients with high IgA levels and infection as secondary diagnoses (17 patients). The sensitivity was calculated to be 58.10%, and the specificity was 74.62%. These findings indicate that the model is modest as a screening tool (the model correctly detects approximately 58.10% of positive cases) and reliable as a confirmatory test (the model accurately identifies approximately 74.62% of true negative cases). The positive predictive value (PPV) was 71.66%, suggesting that more than two-thirds of the patients with a positive screening test had normal IgA levels with an infectious disease. The negative predictive value (NPV) was 61.72%, indicating that nearly two-thirds of patients with a negative screening test (high IgA levels) were free of infectious diseases.

Similar to the ROC curve provided for the IgA serum levels, we also applied the ROC curve to test how well serum IgG and IgM levels could tell the difference between patients who had infectious diseases as a secondary diagnosis and those who did not during the admission phase. The ROC curve analyses were conducted on the same cohort of patients included in the IgA ROC analysis. The ROC curve results indicated that IgG levels achieved statistical significance, with an AUC of 0.864 and a standard error of 0.048, demonstrating a strong discriminative capacity for diagnosing normal IgG levels in patients with infections, as detailed in Table 2 and in the graphic representation presented in Figure 6a. Moreover, the Youden’s index equals 0.638, with a cutoff value for the ROC curve of 13.080 g/L, suggesting that the test has a reasonably good ability to distinguish between positive and negative cases. Similar to IgA serum levels groups, patients with normal IgG levels and infection as secondary clinical diagnoses were classified as true positives (49 patients), whereas those with high IgG levels but no clinical infectious disease diagnoses were classified as false negatives (9 patients). True negatives included patients with normal IgG levels and no infection as secondary clinical diagnoses (71 patients), and false positives comprised patients with high IgG levels and infection as secondary diagnoses (12 patients). The sensitivity was calculated to be 84.50%, and the specificity was 85.5%. These findings indicate that the model is highly effective as a screening tool and as a confirmatory test. The PPV was 80.30%, indicating a high probability that a positive test result accurately reflects the presence of the condition. The NPV was 88.80%, indicating that almost all patients with a negative screening test (high IgG levels) were free of infectious diseases. Regarding the ROC curve provided for the IgM serum levels, the results indicated that IgM levels achieved statistical significance, with an AUC of 0.663 and a standard error of 0.053, demonstrating a very low capacity for diagnosing normal IgM levels in patients with infections, as detailed in Table 2 and in the graphic representation presented in Figure 6b. Moreover, the Youden’s index equals 0.353, with a cutoff value for the ROC curve of 1.145 g/L, suggesting that the test has some ability to distinguish between positive and negative cases, but it is not highly reliable.

Among all the patients enrolled in the study, 26% (37 patients) had a recent history of, or an infection associated with, Group A beta-hemolytic streptococcus. Additionally, apart from infections as secondary diagnoses, patients also exhibited persistent nephropathy (6.3%) and inflammatory bowel disease (4.9%).

### 3.4. Paraclinical Results

In all patients included in the study, the initial symptom at the onset of HSP was the purpuric rash. Regarding the initial rash localization, the lower limb was the most common site of purpuric eruption (84.61%). The cutaneous signs were most typically found on the calves (84.81%), thighs (65.03%), ankles (67.13%), and foot (67.13%). Notably, only 60.13% of patients (86 people) had cutaneous eruptions over their entire lower limb. The gluteal region (24.47%) and upper limb (15.37%) were the subsequent most afflicted sites for purpura in individuals with a secondary diagnosis of infectious disease. There were 14 patients (9.79%) who presented with an unusually confined purpuric eruption onset. These individuals developed IgAV in the left ear (two patients), thorax (one patient), right wrist joint and left arm (two patients), and upper abdomen (two patients). Two individuals had generalized HSP localization, whereas five others had the eruption on their scalp, submentum, or face. The most frequent combination of IgAV manifestations was observed on the lower and upper limbs (15 patients, 10.48%) and on the lower limbs and gluteal area (24 patients, 16.78%), as detailed in Table 3. Approximately one-third of patients (46 patients) had relapses, with 26.6% experiencing moderate symptoms and 58.3% suffering from severe symptoms. The average number of relapses was 1.5 episodes (range: 1 to 5), necessitating hospitalization exclusively for patients with a secondary diagnosis of infectious disease. The mean duration from the initial onset of the eruption to hospitalization was 3.7 days, with a range of 0 to 16 days.

Aside from purpura, the most common clinical symptoms were joint involvement (76%), followed by digestive issues (48%). Renal impairment occurred in 13.9% of patients, followed by testicular involvement with acute orchepididymitis in 8.5% of cases. Severe forms of the disease more frequently presented thrombocytosis (30.55% of all patients and 48.80% of severe cases) and hypoalbuminemia (11.11% of all patients and 17.02% of severe cases) at admission, while leukocytosis (47.22% of all patients) at admission was associated with renal impairment. Meanwhile, those who had elevated IgA levels exhibited edema and renal impairment. All patients with a secondary diagnosis of infectious disease and severe HSP exhibited renal impairment.

Regarding the administered treatment, antihistamine therapy was utilized in 142 patients (99.30%), and nonsteroidal anti-inflammatory drugs (NSAIDs) were used in 49 patients (34.26%) as the primary approach in supportive care. Corticosteroid therapy was administered to 113 patients (79.02%) with moderate severity to reduce inflammation and alleviate purpuric manifestations. Immunological therapy was required in only 3 patients, who presented with significant renal involvement. For all administered active substances, the dosage was determined based on each patient’s age and weight, in accordance with the specifications provided in the product insert.

In addition, 78 patients (54.54%) were treated with antibiotics after being identified with bacterial infections. All patients with a subsequent diagnosis of infectious disease received antibiotic treatment, either for the related bacterial infection or for bacterial infections acquired as complications during hospitalization. The mean duration of antibiotic administration was 7 days, with a minimum of 3 days for Ceftriaxone and a maximum of 21 days for Meropenem. In the study cohort without a secondary diagnosis of infectious disease, the most frequently administered antibiotics were Azithromycin (33.7%) and Cefuroxime (27.7%). In contrast, for the cohort with a secondary diagnosis of infectious disease, Ceftriaxone (32.6%) was the most commonly administered antibiotic. All antibiotics were administered following the provision of the antibiogram. The antibiotic dosage was administered in accordance with each patient’s age and weight, adhering to the specifications provided in the product insert.

## 4. Discussion

### 4.1. Demographic Characteristics Correlated with HSP Manifestations

The current etiopathogenic paradigm for IgAV involves an aberrant immune response produced by a variety of antigenic external stimuli in genetically vulnerable individuals [7,38]. Given the average age of our study group, we may conclude that IgAV primarily affects children under the age of ten in over 70% of cases, with the peak incidence occurring between the ages of four and seven [39]. This age interval relates to the time when children start school or kindergarten and are exposed to communicable diseases [39]. Furthermore, our statistics indicate that the onset of HSP occurred throughout the year, with the highest prevalence in autumn. This period coincides with the start of the school year and the increased socialization of children. Additionally, consistent with previous research, we observed a lower incidence of IgAV during the summer months, which correlates with a reduced incidence of respiratory viral infections and decreased bacterial transmission during school closures [40,41,42]. Moreover, no patients with mild-severity symptoms were admitted during this period; the majority of hospitalized patients were those experiencing relapses. The interaction of environmental factors, such as urban and rural settings, is considered a significant determinant in the formation of HSP. This interaction reflects epidemiologic disparities among ethnicities, instances of familial aggregation, and interindividual variability in disease prognosis [43,44]. According to the study results, the majority of patients who developed HSP were from urban areas, with a 50% higher prevalence compared to rural areas. Within the study cohort, a significant proportion, exceeding two-thirds, exhibited moderate disease status. This was particularly notable in the urban patient group, where the prevalence was nearly double that of the rural population. The higher population density in urban areas increases the risk of contracting infectious diseases [45]. When comparing genders, males seem to be more susceptible than females; according to the study, the male cohort was almost two times bigger than the female cohort [46]. All in all, the study emphasizes that urban living and educational settings substantially affect the occurrence and severity of IgA vasculitis in children, exhibiting a pronounced peak in the autumn and a greater prevalence in males.

### 4.2. Relationship of Immune Parameters in Children with Henoch–Schönlein Purpura and Infectious Diseases

As mentioned before, plasma cells in mucosa-associated lymphoid tissue (MALT) produce IgA, which is found in the nasopharynx, tonsils, and gastrointestinal mucosa. IgA–antibody immune complexes are formed in response to antigenic exposure from an infection or medication [47]. The study examined the immunological response during infectious diseases, comparing patients who tested positive for various infections with those who tested negative. The immunological response varied significantly between the two phases, underscoring the impact of the time interval and infectious disorders on immunoglobulin levels. Based on our findings, serum IgA levels appear to be raised in almost 40% of HSP patients, particularly those with moderate or severe manifestations, and are prone to be found in cohort without infectious disease as a secondary diagnosis [22]. The study results indicate that IgA serum levels typically remain within the normal range when patients are hospitalized with an infectious disease as a secondary diagnosis. The study results indicate that IgA serum levels typically remain within the normal range when patients are hospitalized with an infectious disease as a secondary diagnosis [48]. It is important to emphasize that IgA serum levels, albeit being reduced post-treatment, persisted at or over the upper limit of normal in individuals with a subsequent diagnosis of infectious disease. In contrast, the cohort that tested negative for infectious diseases exhibited completely normalized IgA serum levels post-treatment [22]. According to the study results, infectious diseases prolong elevated IgA serum levels and increase the probability of relapses [49]. The study indicates that serum IgA levels have significant discriminative ability for diagnosing normal IgA levels in patients with infectious illnesses, as shown by the ROC curve analysis. This analysis supports the use of serum IgA levels as a reliable confirmatory test for diagnosing normal IgA levels in such patients. The study underscores the importance of utilizing serum IgA levels in conjunction with other diagnostic tools to enhance the accuracy of diagnosing infectious diseases in pediatric patients [7,24,37]. Notably, although the IgA serum levels were in the normal range for the patients with infectious disease as a secondary diagnosis, the symptoms presented moderate manifestations. According to specialized literature, HSP is associated with a variety of childhood infections and genetic variables [48,50]. This study aligns with the hypothesis that HSP may be precipitated by a series of viral infections, exposure to certain medications, insect bites, immunizations, food allergies, and other factors. This study supports this hypothesis [48,50]. Serum IgA levels are a vital biomarker for differentiating the immune response in IgA vasculitis, underscoring the considerable influence of infectious diseases on disease development and relapse rates.

Elevated IgG and IgM levels in pediatric HSP patients highlight the significant role of infectious diseases in modulating immune responses and disease severity. Elevated IgG and IgM levels in pediatric HSP patients are strongly linked to infectious diseases [51]. Moreover, patients who tested positive for pathogens in their IgG- or IgM-specific blood tests also had high IgG or IgM serum levels either during or post-treatment. IgG is the most prevalent antibody in the bloodstream and plays a critical role in the immune response, particularly in conferring long-term immunity and protection against infections [52]. During the early stages of infection, IgG levels begin to increase as the immune system responds to the presence of pathogens, typically peaking after the initial immune response and providing sustained protection against the pathogen [53]. Furthermore, the extent of serum IgG alterations in our pediatric patients was significantly influenced by subsequently diagnosed infectious diseases. The levels remained within the normal range during infection but increased markedly post-infection in patients with a secondary diagnosis of infectious disease. In contrast, patients without a secondary diagnosis of infectious disease maintained normal levels in both phases. Although previous investigations on serum IgG levels in HSP patients have yielded mixed results—some indicating declines, increases, or no significant changes—our study results reveal a substantial difference in IgG levels between patients with and without infections. Moreover, IgG serum levels demonstrate strong diagnostic accuracy for identifying infectious diseases as a secondary diagnosis, highlighting the superior utility of IgG as a biomarker in clinical settings. The immunological panel elucidated the severity and manifestation of the disease [22,54,55].

IgM plays a crucial role in the immune response, particularly during the early stages of infection. IgM is the first antibody produced by the immune system in response to an infection [51]. It acts as an early defense mechanism, quickly identifying and neutralizing pathogens, which is why, during the early stages of infection, IgM levels rise rapidly as the immune system responds to the presence of pathogens [56]. In contrast to IgA and IgG serum levels, the variations in IgM serum levels tend to decrease over time post-treatment, with a nearly equal decrease observed in both groups. Furthermore, in the cohort of patients with a secondary diagnosis of infectious disease, IgM serum levels tend to be directly proportional to IgA serum levels. All in all, IgM levels surge as the first line of defense during early infection, highlighting its pivotal role in the immune response and its correlation with IgA levels in patients with secondary infectious diseases.

### 4.3. The Correlation Between Infectious Diseases and HSP Manifestations and Treatments

Respiratory infections, especially those induced by Group A β-hemolytic streptococcus, are common in individuals with IgA vasculitis, highlighting the critical influence of upper respiratory tract infections on the initiation and advancement of the condition. In this research and prior publications, symptoms of infection, primarily in the upper respiratory tract, were observed in the majority of cases within days after the onset of IgAV symptoms [16,41]. The study group in the present study had a high prevalence of respiratory infectious illnesses, with β-hemolytic streptococcus being the primary pathogen. Group A β-hemolytic streptococcus is the most investigated pathogen associated with HSP, as it can be found in up to 50% of persons with acute HSP, either by serological testing or bacterial cultures [57,58].

This study’s findings offer substantial insights into the clinical manifestation, development, and treatment of HSP in pediatric patients. The initial symptom of HSP in all patients was a purpuric rash, predominantly localized to the lower limbs, particularly the calves, thighs, ankles, and feet. This localization pattern aligns with the existing literature, which often cites the lower extremities as the primary site of purpuric eruptions in HSP patients [59,60]. The study also highlights the variability in rash localization, with some patients presenting with purpura in less common areas such as the gluteal region and upper limbs and even confined to specific sites such as the ear or thorax [22]. A significant observation is the elevated relapse incidence among patients, especially those with a concomitant diagnosis of infectious disease. Purpuric rashes in pediatric HSP patients predominantly affect the lower limbs, with infection-related relapses highlighting the critical need for vigilant infection management. The average number of relapses and the necessity for hospitalization in these cases emphasize the importance of monitoring and managing infections in HSP patients to prevent recurrent episodes [46].

Joint involvement and digestive issues were the most common clinical symptoms aside from purpura, with renal impairment and testicular involvement also observed during the study. Gastrointestinal signs, which occur several days or even a week before cutaneous symptoms, might cause clinical perplexity until the rash appears; in severe cases, it requires immediate immunosuppressive treatment [59]. The correlation between elevated IgA levels and renal impairment further supports the role of immunological factors in the pathogenesis of HSP [25]. The clinical presentation of HSP is significantly compounded by musculoskeletal and intestinal problems, as well as renal impairment. They also show how important immunological factors are in its development.

The treatment regimens used in this trial, which included antihistamines, NSAIDs, corticosteroids, and immunological therapy, are current therapeutic practices [22,60]. The increased use of antibiotics in patients with secondary bacterial infections emphasizes the need of treating infectious complications in HSP therapy [41]. The difference in antibiotic choice across cohorts with and without secondary infections highlights the importance of individualized therapeutic methods based on individual patient characteristics and infection status [8]. Customized therapy protocols, incorporating antibiotics for secondary infections, are essential in the management of HSP, highlighting the significance of personalized therapeutic strategies.

As a key limitation, because this is a retrospective study, IgA levels were evaluated only once during hospital admission after clinical signs occurred and once post-treatment during reevaluation. Thus, increased or normal IgA levels in patients presenting various types of HSP manifestations may have been impacted by recurring infections. To determine this connection, IgA levels could be evaluated prior to, during, and following infections and monitored for a longer period of time. Moreover, this study was conducted in a pediatric clinic and consisted of an extensive evaluation of clinical observation sheets. The study did not involve an examination of IgA subcategories or socio-economic status due to the fact that the information was not previously recorded.

## 5. Conclusions

The study underscores the importance of analyzing serum IgA levels in conjunction with other diagnostic tools to enhance the accuracy of diagnosing infectious diseases in pediatric patients with HSP. The statistical results showed that there is a correlation between the frequency of infectious diseases and high IgA serum levels in patients diagnosed with HSP. Furthermore, IgAV primarily affects young children; over 70% of cases occur in children under ten, with peak incidence between ages four and seven, coinciding with school or kindergarten exposure to communicable diseases. Thus, infectious diseases play a crucial role in modulating immune responses and disease severity in pediatric HSP patients, with normal IgG and elevated IgM levels highlighting this impact. Moreover, elevated IgG and IgM levels in pediatric HSP patients underscore the pivotal role of infectious diseases in shaping immune responses and disease severity, highlighting IgG’s superior diagnostic accuracy as a biomarker in clinical settings.

## Figures and Tables

**Figure 1 biomedicines-13-00113-f001:**
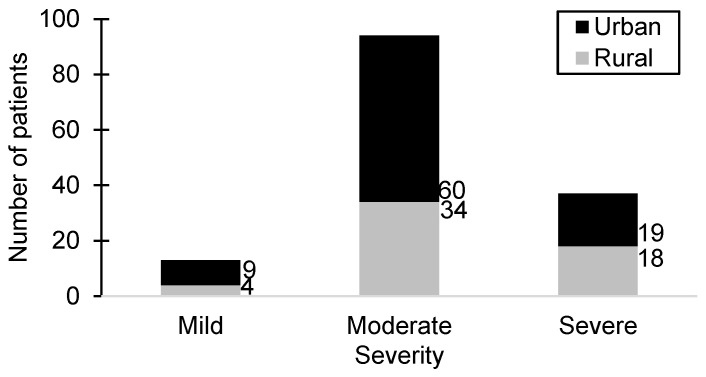
The distribution of patients according to HSP severity level and environment.

**Figure 2 biomedicines-13-00113-f002:**
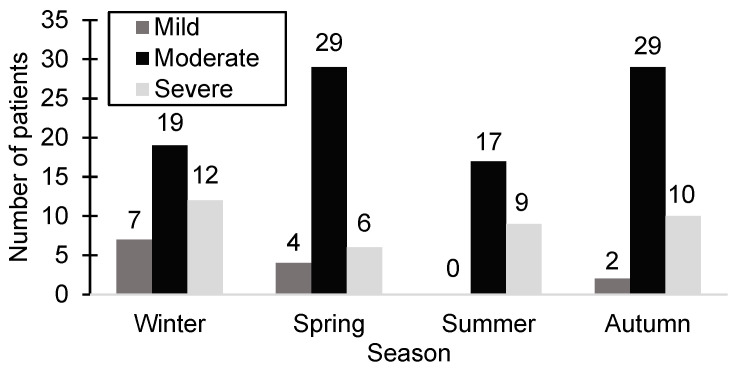
The distribution of patients according to season and the severity of HSP.

**Figure 3 biomedicines-13-00113-f003:**
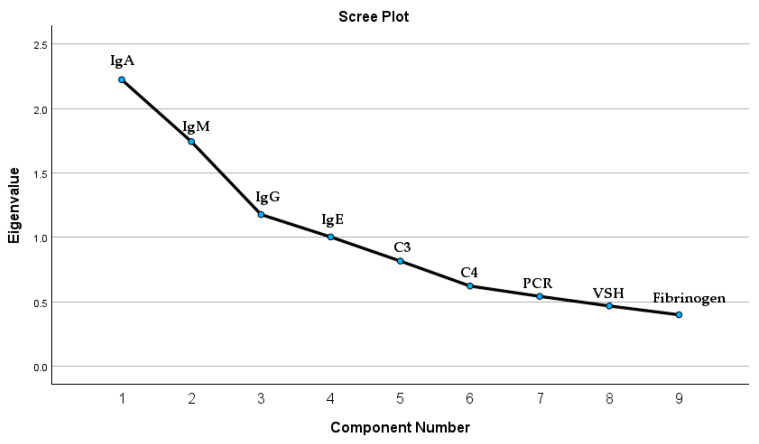
The graphic representation of the PCA test results for possible correlations.

**Figure 4 biomedicines-13-00113-f004:**
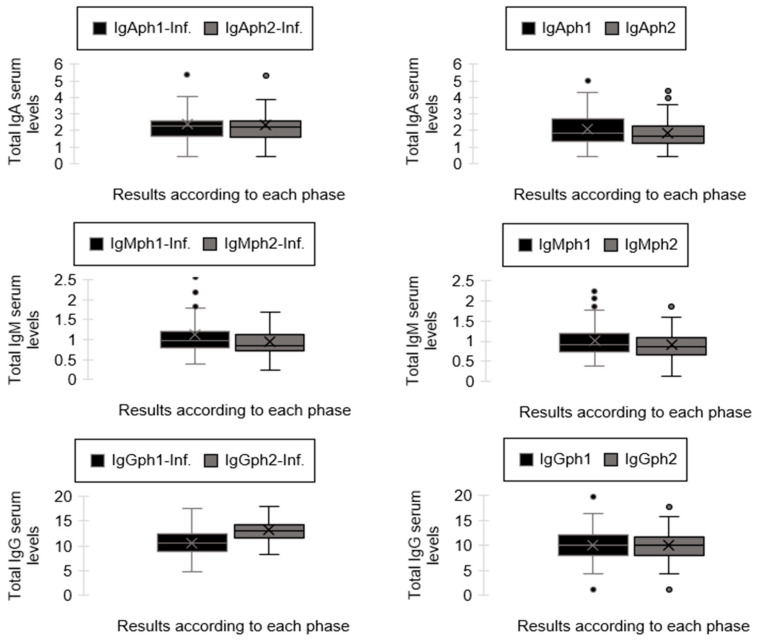
The graphical representation of IgA, IgM, and IgG serum levels (g/L) for both patient groups across both phases. The circles represent the outliers.

**Figure 5 biomedicines-13-00113-f005:**
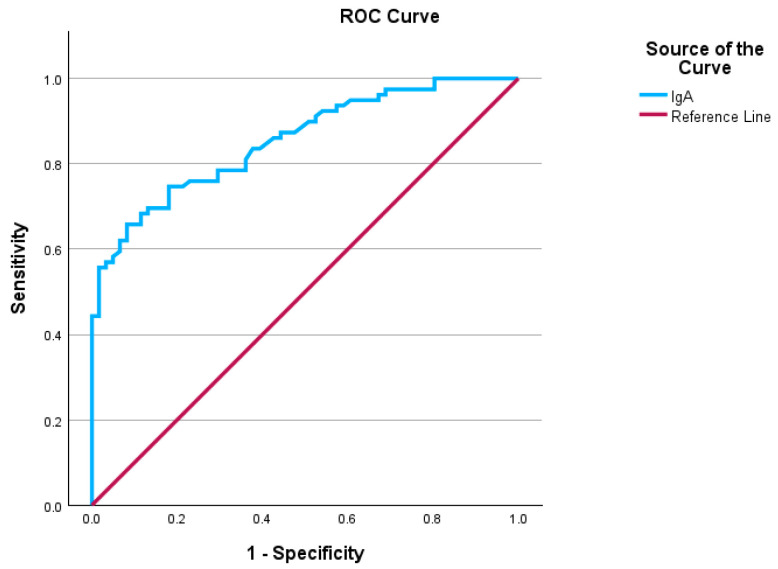
ROC curves obtained from the IgA serum levels. Sensitivity is shown in the ordinate, while the false positive rate (1-specificity) is presented in the abscissa.

**Figure 6 biomedicines-13-00113-f006:**
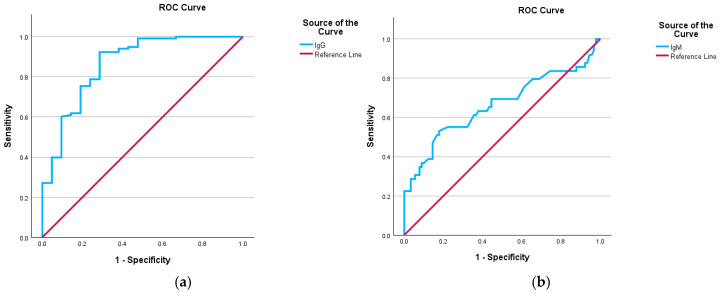
(**a**) ROC curves obtained from the IgG serum levels. (**b**) ROC curves obtained from the IgM serum levels. For both figures, sensitivity is shown in the ordinate, while the false positive rate (1-specificity) is presented in the abscissa.

**Table 1 biomedicines-13-00113-t001:** ROC analyses for total IgA levels (area under the ROC curve and test result variable(s)).

Area	Std. Error ^a^	Asymptotic Sig. ^b^	Asymptotic 95% Confidence Interval	
			Lower Bound	Upper Bound
0.855	0.031	0.000	0.795	0.915

^a^. Under the nonparametric assumption. ^b^. Null hypothesis: true area = 0.5.

**Table 2 biomedicines-13-00113-t002:** ROC analyses for total IgG and IgM serum levels (area under the ROC curve and test result variable(s)).

Parameter	Area	Std. Error ^a^	Asymptotic Sig. ^b^	Asymptotic 95% Confidence Interval	
				Lower Bound	Upper Bound
IgG	0.864	0.048	0.000	0.770	0.958
IgM	0.663	0.053	0.002	0.559	0.767

^a^. Under the nonparametric assumption. ^b^. Null hypothesis: true area = 0.5.

**Table 3 biomedicines-13-00113-t003:** The initial cutaneous manifestations in patients with HSP.

Initial Localization	Number of Patients
calves	122
thighs	94
ankles	97
foot	97
gluteal region	35
upper limbs	22
ear	2
thorax	1
abdomen	2
scalp/face	5

## Data Availability

The data presented in this study are available on request from the corresponding author. The data are not publicly available due to restrictions such as privacy and ethics.

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
