# Peer review of "The Impact of Infectious Diseases on Clinical Characteristics and Immunological Correlations in Pediatric Henoch–Schönlein Purpura: A Five-Year Retrospective Study"

_biomedicines, 2025, doi:10.3390/biomedicines13010113_

Round 1
Reviewer 1 Report (Previous Reviewer 2)
Comments and Suggestions for Authors
This is a retrospective cohort study of 144 patients with IgA vasculitis (IgA-V) to evaluate the association with infectious diseases. The presented manuscript is described in a reasonable manner, but I do not understand the method of statistical analysis that the authors performed to evaluate whether clinical indicators differ in IgA vasculitis associated with or without infection. Rather than dividing the groups into infection-related and non-related groups and comparing clinical indicators (such as serum immunoglobulin levels) at the time of onset and at the time of remission, shouldn't authors divide them into the states at the time of onset and at the time of remission and compare clinical indicators between the infection-related and non-related groups? In addition, please provide the units (mg/dL, etc.) of serum IgA, IgG, and IgM levels in Figure 4 and Table A2.
Authors have assessed the diagnostic performance of serum IgA levels using receiver operating characteristic (ROC) curves to distinguish between patients who experienced infection as a secondary diagnosis during hospitalization and those who did not, but what about other clinical indicators (such as serum IgG, IgM, IgE, C3 and C4 levels)? In addition, the cut-off level for serum IgA in the ROC curve was described to be 2.0000, but what was the unit?
Author Response
Please see the attachment.

Reviewer 2 Report (Previous Reviewer 3)
Comments and Suggestions for Authors
· Overall, the handed-in revision of the study appears to have improved the overall quality of the manuscript slightly. Throughout the entire manuscript there are still wordings that prevent the reader’s attention to develop into a positive direction and the overemphasis on the most simple statistical aspects (test for normal distribution, explaining in three lines -L152-154- why the median and not the mean was utilized), the ineffective data visualization and - most importantly – the still missing carved out punch line/ unmet need make the manuscript a truly tough one to read. So, the reader really has to have a strong intrinsic motivation to get informed on the topic of “IgAV in Romania” to read through the manuscript, because the authors appear to hinder any extrinsic motivation.
· Positive aspects of the revision are that the authors now mention classification criteria (L 84-86) and the ROC curve is nicely shown.
· To mention the limitations of the study in the conclusions part (L. 455-463) underscores my point of criticism that the authors did not succeed in carving out a nice punch line interesting to follow.
· For an international scientific community, characterizing IgAV in a pediatric population still is interesting.
· Therefore, I recommend: Major revision.
Round 2
Reviewer 1 Report (Previous Reviewer 2)
Comments and Suggestions for Authors
This is a retrospective cohort study of 144 patients with IgA vasculitis (IgA-V) to evaluate the association with infectious diseases. The presented study was well performed and reanalyzed. The revised manuscript is described in a reasonable manner. Authors had also responded to my all comments. However, the serum immunoglobulin levels (especially the units) appear to be incorrect. If those levels are too different, it will affect the reliability of this study, so please check and correct them.
Author Response
Please see the attachment.

This manuscript is a resubmission of an earlier submission. The following is a list of the peer review reports and author responses from that submission.
Round 1
Reviewer 1 Report
Comments and Suggestions for Authors
The manuscript by OpriÈ›escu et al. presents a retrospective study aimed at investigating the clinical features of children diagnosed with IgA vasculitis/Henoch-Schönlein purpura (IgAV/HSP) and the role of infectious diseases in this population. The study touches upon an important topic, as understanding the relationship between infections and IgAV is crucial for improving clinical management. However, there are several areas that could be improved to enhance the contribution of this work.
Major Comments:
While the study addresses an established area of research, one of the main concerns is the limited novelty of the findings. The association between IgAV/HSP and infectious diseases, as well as the clinical features presented, have been well-documented in previous studies. It would be helpful for the authors to clarify how their work builds upon or provides new insights compared to the existing literature.
Another limitation is the sample size and the generalizability of the results. Although the study includes 144 cases, the findings may not be widely applicable, especially if the sample was drawn from a single hospital or region. This could limit the study's relevance to broader pediatric populations. Expanding the sample size or conducting a multicenter study could enhance the robustness of the conclusions.
The retrospective nature of the study also presents challenges in establishing causal relationships or observing disease progression over time. A prospective or longitudinal design would allow for better tracking of changes in immunological markers and disease outcomes, providing more insight into the temporal relationship between infections and the onset or severity of HSP. Additionally, addressing how the immune response changes over time could strengthen the conclusions about immunoglobulin levels and their correlation with infectious diseases.
Finally, it would be valuable for the authors to consider potential confounding factors that may influence both IgA levels and HSP manifestations. Factors such as environmental influences or other immunological markers could play a role, and it would be beneficial to control for these in the analysis. Including stratified or adjusted analyses based on demographic variables would also add to the strength and reliability of the findings.
Minor Comments:
Lines 118-119: It is important to note that the EULAR/PRINTO/PRES criteria are classification criteria rather than diagnostic criteria, so this should be clarified.
Lines 202 and 219: The authors mention "severe forms of the disease" and "severe symptoms" without defining how disease severity was graded. Providing a clear definition of what was considered more severe symptoms would help readers interpret the findings more accurately.
Nephritis: It would be useful to include more detailed information regarding patients with nephritis, such as urine findings, biopsy results, and whether the patients experienced nephritic/nephrotic syndrome or chronic kidney disease.
Therapy Details: Additional information on the therapies used, particularly doses of glucocorticoids and the duration of their administration, would provide better insight into treatment approaches and their outcomes.
Reviewer 2 Report
Comments and Suggestions for Authors
This is a retrospective cohort study of 144 patients with IgA vasculitis (IgA-V) to evaluate the association with infectious diseases. The presented manuscript is described in a reasonable manner, but the analysis results seem to be insufficient in my opinion. To evaluate the association between IgA-V and infectious diseases, patients with IgA-V should be divided into those with and without infection and various clinical indicators should be compared.
Moreover, there are several points that I would like to inquire as follows.
1. The results of the ROC analysis were not reflected in the abstract.
2. I don't understand the purpose of looking at the correlation between several immunological tests. Rather, I think authors should evaluate the relationship between each immunological indicator and clinical symptoms, severity, prognosis, etc. For example, did serum IgA levels correlate with disease severity or any clinical symptoms?
3. Please provide the cut-off value for serum IgA level in the ROC analysis.
4. Because the sensitivity and specificity were relative low, I do not think that serum IgA levels were useful as a biomarker for infectious diseases. What about other parameters? Would it be possible to increase specificity and sensitivity by combining multiple parameters?
5. In Figure 1 and 2, ‘Easy’ should be changed to ‘Mild’ in Figure 1 and 2.
6. Please recheck the terminology. (e.g. protein C reactive should be changed to C-reactive protein.)
Reviewer 3 Report
Comments and Suggestions for Authors
Reviewer report
biomedicines-3243059
„Correlation between IgA levels in Henoch–Schönlein purpura and a medical history of infectious disease in the pediatric population: a five-year retrospective analysis”
Evaluation
· Overall, the study in it’s current version appears totally mediocre and faint in scientific expressiveness, which is a pity, since there are some interesting aspects about it. The entire manuscript requires major revision, since no punch line is detectable throughout the entire manuscript.
· Therefore, I recommend: Major revision
Major issues
· The weakest points of the study:
o s. “Discussion” + “Conclusion”
Minor issues
Title
· I recommend to use an active title which expresses the main finding. To merely say “correlation” is featureless and bores the reader
Abstract
· p.1L16-19: Please connect the first two sentences and point out the unmet need justifying the study conduction by using “however” in the third sentence. Read in the way it is written right now the introductory part of the abstract, which is one of the most relevant parts of a manuscript, appears somehow pointless. The aim of the third sentence starting with “however” is to lead over to the aims of the manuscript provided in line 19.
· The aims outlined on p.1L19 + L21 need to be compacted: talking about clinical characteristics of children in general in line 19 and then writing “furthermore” in the next sentence eventhough demographic characteristics such as “sex, area, age” are also clinical data appears irritating in the reading flow.
· p.1L25: Please outline that patient blood samples, so serum measurements, were used right in the beginning of the sentence; otherwise, one searches for this decisive information throughout the entire sentence, leading to a sort of reading fatigue.
· p. 1L 29-30: the difference between the two tests must be pointed out in a clearer way, otherwise it is unclear why this finding is worth mentioning at all.
Introduction
· p. 1L36: Please refer to the most relevant classification system: CHCC revised 2012 classifiying IgAV as an immune-complex SVV [1]. This must be mentioned when talking about any vasculitis!!
1. Jennette, J.C.; Falk, R.J.; Bacon, P.A.; Basu, N.; Cid, M.C.; Ferrario, F.; Flores-Suarez, L.F.; Gross, W.L.; Guillevin, L.; Hagen, E.C.; et al. 2012 Revised International Chapel Hill Consensus Conference Nomenclature of Vasculitides. Arthritis & Rheumatism 2013, 65, 1–11, doi:10.1002/art.37715.
· p.1L43/44 + p.2 L51/52: why do the authors paraphrase the same content twice?
· Please shorten the introduction and point out the most relevant aspects that are connected to your findings. The reader is not interested in reading a Wikipedia-like article.
· p. 2L 74-95: this is where the introduction becomes interesting, but the authors appear to ruminate the same aspects over and over again. Why not try to be be succinct?
Methods
· p.3L105-109: please provide the exact technical information of the ELISA kits being used in this study.
· I recommend to position the inclusion and exclusion criteria before the statistics.
· p.3L134: inclusion criteria say: age range between 0 and 18 were included. Now, exclusion criteria say patients below the age of 1y were excluded. Please correct accordingly.
· p.3L138 + p.3L143: Please mention this COVID-19-aspect only once, either in it’s negating or affirmating form
Results
· p.4L151-153: the authors do not seem to be aware of the fact that in case of non-normal distribution the required statistical value has to be median and IQR, not (!) mean and SD. Please correct accordingly.
· P.3L112: Why do the authors call it C-reactive protein in line 112 and “protein C reactive (PCR)” in line180?
· P.5L189: Why do the authors explain basic statistical methods which are ought to be known to a scientific community?
· The results provided solely being based on Pearson’s correlation are really not strong
· P.5L197-206: this overview provided is interesting and presented in a sound way.
· P.6L218: the authors write about “buttocks” in a scientific manuscript, which is ought to be serious??? This must be a joke. Please rephrase with a serious term.
· P.6L228-251: the ROC analysis is the true highlight of this paper. It really should be pointed out more in the abstract and also the results part. Please draw the attention to it: this is how you could polish up this manuscript. I invite the authors to perform a calculation of the Youden index to identify the threshold of IgA levels where dichotomization could be further carried out.
Discussion
· P.7L263-265: please do not repeat the information given in the introduction already: it really fatigues the reader.
· Could you please provide a discussion and not a review? So, where is the reflection on the author’s findings and the embedding into the current body of evidence? I mean, it is interesting and scientifically correct, but this is not the meaning of a discussion. The reader somehow gets lost in an ocean of information and cannot connect it with the findings of the manuscript. So, the punchline of the entire manuscript (in it’s current version) is missing.
Conclusion
· P.9L357: The authors claim that “there is a correlation between the frequency of infectious diseases and normal IgA levels in patients diagnosed with HSP”. But where is the statistically significant finding being presented in the results section for this statement? They provided an interesting ROC analysis, yes, but this from it’s statistical nature is no correlative analysis. And I cannot find the relating results supporting the statement.
Tables
· p.10 Table A2: the visualization of Pearson’s correlation really needs refurbishment. It’s confusing.
Figures
· p. 10, Figure A1: The picture on the left hand side seems to be violating the children’s personal rights by exposing the gluteal region. Please remove this part of the picture.
Comments on the Quality of English Language
fine